# Diagnosis and management of an inappropriate sinus tachycardia in adolescence based upon a Holter ECG: A retrospective analysis of 479 patients

Reiner Buchhorn*[◉], Christoph Baumann, Semanur Gündogdu[◉], Ulla Rakowski, Christian Willaschek[◉]

Department of Paediatrics, Caritas Krankenhaus, Bad Mergentheim, Germany

◉ These authors contributed equally to this work.
* buchrein@gmail.com

**Data Availability Statement:** All relevant data are within the manuscript and its Supporting Information files.

## Abstract

Inappropriate sinus tachycardia (IST) is a common disease of the autonomic nervous system in children and adults. Diagnosis and treatment of IST in adolescents is not well defined. In this retrospective study, we tested our hypothesis regarding autonomic dysfunction in childhood by analyzing 24-h heart rate variability (HRV) in 479 children, with a mean age of $13.7 \pm 2.1$ years, who were referred to the outpatient clinic in the Pediatrics Department within the last 15 years. Seventy-four adolescents with a mean 24-h heart rate $\geq$ 95 bpm (our cut-off for an IST based upon 66 healthy controls) were deemed to have IST. We found the risk of IST to be high in adolescents with attention deficit disorder (OR = 3.5,p<0.001), pre-hypertension (OR = 2.5, p = 0.043) and hypertension (OR = 2.1,p = 0.02); insignificantly enhanced in children with short stature (OR = 1.9,p = 0.19), surgically-treated congenital heart disease (OR = 1.4,p = 0.51) and obesity without hypertension (OR = 1.4;p = 0.25); and negligible in adolescents with anorexia nervosa (OR = 0.3, p = 0.26) and constitutional thinness (OR = 0.9,p = 0.89). IST was associated with a significant decrease in global HRV and elevated blood pressures, indicating an enhanced cardiovascular risk. Methylphenidate did not increase 24-h heart rates, whereas omega-3 fatty acid supplementation significantly decreased elevated heart rates and increased HRV in adolescents with IST. In this retrospective analysis, 15.4% of adolescents suffered from IST with a 24-h heart rate $\geq$ 95 bpm, predominately due to attention deficit disorder and hypertension.

## Introduction

Postural tachycardia syndrome, inappropriate sinus tachycardia (IST) and vasovagal syncope are common diseases of the autonomic nervous system in children and adults. Working criteria for diagnoses and recommendations on their assessment and management were published in a 2015 paper entitled "Heart Rhythm Society Consensus Statement on the Diagnosis and Treatment of Postural Tachycardia Syndrome, Inappropriate Sinus Tachycardia, and

**Funding:** We thank the Blaschek foundation who supports us by covering the publication fees for this research project. The funders had no role in study design, data collection and analysis, decision to publish, or preparation of the manuscript.

**Competing interests:** The authors have declared that no competing interests exist.

Vasovagal Syncope" [1]. Representatives of the Pediatric and Congenital Electrophysiology Society were engaged in this paper; however, the incidence, diagnostic criteria, and pharmaco-therapy of IST in children remain ill-defined. Most patients develop the first clinical symptoms as adolescents during maturation of their autonomic nervous system. Hence adolescence appears to be a vulnerable phase in the development of these diseases and offers a potential window of opportunity for therapeutic interventions.

Over the last 20 years, we have used Holter ECG monitoring as a diagnostic tool to analyze autonomic dysfunction and monitor pharmacotherapy in childhood. Our publications focused on basic pathophysiology and new therapeutic approaches:

1. **Heart Failure:** Beta blocker therapy for heart failure in infants, children, and adults with congenital heart disease [2].

2. **Early Life Stress:** The impact of early life stress on height and neurodevelopmental impairment [3].

3. **Attention Deficit Disorder**: Stress and cognitive dysfunction in children [4].

4. **Nutritional Disorders:** The impact of nutrition on the autonomic nervous system [5].

5. **Omega-3 fatty acid supplementation:** Effects on heart rate variability (HRV) in children with autonomic dysfunction [6].

In the current analysis, we test the hypothesis that an elevated mean 24-h heart rate, known as inappropriate sinus tachycardia, is the common final pathway of a group of children with autonomic dysfunction who had a high cardiovascular risk in later life. We calculate the risk of IST in eight patient groups and the effect of pharmacotherapy on this risk.

Swedish register data clearly show that an elevated resting heart rate in adolescence is an important cardiovascular risk factor for later all-cause mortality, most commonly due to the higher risk of heart failure [7]. We attempt to differentiate the effect of an elevated heart rate from elevated blood pressures. Finally, we test whether omega-3-fatty acid supplementation effectively reduces the mean 24-h heart rates of children with inappropriate sinus tachycardia.

## Materials and methods

### Patients

Retrospective analysis of patient data was approved by the ethical board of our state's medical chamber (Landesärztekammer Baden Württemberg). First Data of this retrospective analysis were recently published [8]. All patients undergoing any kind of treatment, phamacologic as stimulant therapy or nutritional intervention as omega-3 fatty acid supplementation gave oral informed consent for data analysis, which was documented in the patient charts. Equally oral informed consent was achieved from patients with inappropriate sinus tachycardia. Of course data analysis was skipped if patient consent was denied.

Based upon our hypotheses about autonomic dysfunction in childhood, we analyzed 24-h HRV in 479 children with a mean age of 13.7 ± 2.1 years, who were referred to the outpatient clinic in the Pediatrics Department of the Caritas Hospital, in Bad Mergentheim, Germany, between 2005 and 2019.

For this analysis, we included the initial Holter ECG of each patient, in cases where many of our therapeutic decisions were based upon HRV monitoring as a target of pharmacotherapy and nutritional interventions. From available data of first visits, most patients had no pharma-cotherapy, whereas some patients in groups 1/2/4 had received long-term pharmacotherapy. For further analysis, we stepwise excluded groups who had a known impact on the autonomic

nervous system (Surgically-treated Congenital Heart Disease>Attention Deficit Disorder with/without Hyperactivity>Anorexia Nervosa>Short Stature>Constitutional Thinness>Obesity>Hypertension>Pre Hypertension>Healthy Control). After hierarchical exclusion of these patient groups, a group of 66 children remains that should be autonomic healthy to the best of our knowledge. These data came from patients who attended our outpatient clinic for exclusion of cardiac arrhythmia or heart defects. For descriptive statistics as shown in Tables 1 and 2 we used the following groups:

1. **Surgically-treated Congenital Heart Disease** (N = 55). Pharmacotherapy: Beta blocker = 7, Angiotensin-converting enzyme inhibitor and Angiotensin receptor antagonists = 4, Digoxin = 3, Calcium Antagonist = 1.

2. **Attention Deficit Disorder with/without Hyperactivity** (N = 86). Pharmacotherapy: Methylphenidate = 69, Atomoxetine = 7, Amphetamine = 3.

3. **Anorexia Nervosa** (N = 34). No pharmacotherapy.

4. **Short Stature** (N = 15). Pharmacotherapy: Growth Hormone = 3, Beta blocker = 1.

5. **Constitutional Thinness** (N = 20). No pharmacotherapy.

6. **Obesity** (N = 130). Pharmacotherapy: Beta blocker = 1, Angiotensin-converting enzyme-inhibitor = 1.

7. **Hypertension** (N = 53). No pharmacotherapy at baseline.

8. **Pre-hypertension** (N = 13) No pharmacotherapy.

9. **Healthy Control** (N = 66) No pharmacotherapy.

Seven otherwise healthy children had an IST due to different somatoform disorders. These children could not be assigned to any of the groups and were not included in the descriptive statistics but were included in the omega-3 fatty acid supplementation analysis.

**Table 1. Anthropometric measurements and patient groups.**

| | N | Age [Yrs] | Height [cm] | Height Perc. [%] | Weight [kg] | BMI [kg/m2] | BMI Perc. [%] | Syst. BP [mmHg] | Syst. BP Perc [%] | Diast. BP [mmHg] | Diast. BP Perc. [%] |
|---|---|---|---|---|---|---|---|---|---|---|---|
| **Healthy control** | 66 | 13.8 ± 2.3 | 161.4 ± 11.7 | 55.3 ± 26.5 | 51.5 ± 11.5 | 19.5 ± 2.7 | 49.4 ± 27.5 | 111.2 ± 9.4 | 54.0 ± 25.8 | 62.5 ± 8.1 | 43.6 ± 22.2 |
| **Congenital heart defect** | 55 | 13.9 ± 2.3 | 155.7 ± 14.8 | 39.5 ± 32.8 | 50.8 ± 18.3 | 20.5 ± 5.2 | 52.2 ± 31.9 | 117.0 ± 13.5 | 68.2 ± 3 0.4 | 60.7 ± 10.7 | 40.6 ± 28.7 |
| **Attention deficit disorder** | 86 | 13.2 ± 2.0 | 157.4 ± 12.2 | 43.7 ± 28.6 | 52.2 ± 17.4 | 20.7 ± 5.3 | 54.3 ± 33.6 | 121.1 ± 13.6a | 76.1 ± 25.6a | 65.2 ± 11.1 | 53.3 ± 27.6 |
| **Anorexia nervosa** | 34 | 14.7 ± 2.2 | 163.6 ± 8.1 | 49.1 ± 28.5 | 39.4 ± 5.7a | 14.6 ± 1.3a | 2.3 ± 5.8a | 101.8 ± 11.1a | 29.6 ± 27.8a | 61.4 ± 8.6 | 37.9 ± 23.9 |
| **Short stature** | 15 | 14.5 ± 1.9 | 148.8 ± 8.6a | 8.4 ± 15.9a | 52.0 ± 21.5 | 23.5 ± 10.5 | 57.9 ± 39.8 | 118.4 ± 17.8 | 72.9 ± 27.6 | 64.08.6 | 53.5 ± 21.0 |
| **Obesity** | 130 | 13.3 ± 2.1 | 161.8 ± 10.5 | 62.5 ± 28.7 | 82.2 ± 20.1a | 31.1 ± 5.0a | 98.8 ± 1.4a | 128.8 ± 13.7a | 86.9 ± 18.8a | 64.2 ± 12.8 | 48.5 ± 30.7 |
| **Constitutional thinness** | 20 | 13.8 ± 1.7 | 160.8 ± 10.9 | 46.2 ± 25.3 | 38.7 ± 7.0 | 14.8 ± 11.8a | 1.9 ± 1.7a | 108.2 ± 11.5 | 45.6 ± 35.3 | 59.4 ± 7.7 | 36.2 ± 25.0 |
| **Hypertension** | 53 | 13.8 ± 2.0 | 163.0 ± 12.1 | 52.6 ± 22.7 | 56.3 ± 13.9 | 20.9 ± 2.9 | 63.3 ± 26.1 | 139.1 ± 11.0a | 98.6 ± 1.3a | 69.7 ± 10.2a | 63.8 ± 26.4a |
| **Pre-hypertension** | 13 | 14.2 ± 2.0 | 162.7 ± 9.8 | 45.9 ± 31.3 | 54.8 ± 11.6 | 20.5 ± 2.8 | 57.1 ± 29.7 | 126.4 ± 4.2a | 92.7 ± 1.6a | 72.5 ± 7.3a | 72.1 ± 19.9a |

Unpaired t-test between healthy control and eight patient groups

BMI = body mass index; Perc. = percentile; BP = blood pressure.

[a]P-value < 0.001.

Table 2. Patient groups and heart rate variability.

| | HR | HR day | HR night | SDNN | RMSSD | pNN50 | TP 24 h | VLF 24 h | LF 24 h | HF 24 h | HF/LF 24 h |
|---|---|---|---|---|---|---|---|---|---|---|---|
| | Mean ± SD | Mean ± SD | Mean ± SD | Mean ± SD | Mean ± SD | Mean ± SD | Mean ± SD | Mean ± SD | Mean ± SD | Mean ± SD | Mean ± SD |
| **Healthy control** | 78.2 ± 8.4 | 87.5 ± 10.1 | 66.2 ± 8.7 | 166.9 ± 51.9 | 49.2 ± 15.4 | 28.6 ± 13.2 | 6352 ± 3527 | 3475 ± 2529 | 1709 ± 741 | 1078 ± 888 | 0.61 ± 0.27 |
| **Congenital heart defect** | 81.1 ± 12.4 | 87.8 ± 12.9 | 70.5 ± 13.4 | 144.8 ± 53.5 | 39.2 ± 18.4a | 19.5 ± 14.9a | 5417 ± 4843 | 3209 ± 3521 | 1287 ± 913 | 825 ± 1271 | 0.59 ± 0.36 |
| **Attention deficit disorder** | 87.9 ± 9.5c | 98.3 ± 10.1c | 71.6 ± 13.6 | 153.6 ± 42.6 | 36.1 ± 13.4c | 17.9 ± 12.4c | 4573 ± 2872 | 2670 ± 1995 | 1239 ± 692b | 572 ± 363a | 0.48 ± 0.23 |
| **Anorexia nervosa** | 64.1 ± 12.9c | 72.5 ± 15.9c | 49.4 ± 9.49c | 252.9 ± 70.4c | 62.5 ± 19.3c | 38.2 ± 14.9a | 7933 ± 3587 | 4748 ± 2722 | 1905 ± 802 | 1235 ± 511 | 0.7 ± 0.24 |
| **Short stature** | 87.9 ± 8.9 | 93.5 ± 7.3a | 72.7 ± 9.1 | 130.1 ± 48.4 | 32.6 ± 12.7c | 16.6 ± 10.5 | 4585 ± 2812 | 2512 ± 1726 | 1251 ± 717 | 722 ± 721 | 0.54 ± 0.29 |
| **Obesity** | 85.5 ± 9.4c | 93.9 ± 9.9 | 74.1 ± 10.1c | 138.1 ± 38.4b | 39.8 ± 13.9c | 20.2 ± 12.4c | 4975 ± 3562 | 2703 ± 2003 | 1277 ± 765b | 863 ± 1157 | 0.63 ± 0.36 |
| **Constitutional thinness** | 83.2 ± 8.2 | 90.4 ± 8.8 | 70.6 ± 10.1 | 152.8 ± 35.9 | 39.6 ± 13.5 | 20.1 ± 11.5 | 5668 ± 2825 | 3169 ± 2204 | 1468 ± 622 | 865 ± 548 | 0.63 ± 0.41 |
| **Hypertension** | 81.7 ± 12.4 | 91.4 ± 12.2 | 70.2 ± 11.8 | 156.8 ± 53.83 | 42.9 ± 16.4 | 23.0 ± 13.0 | 6170 ± 4455 | 3829 ± 3518 | 1546 ± 795 | 719 ± 343 | 0.48 ± 0.17 |
| **Pre-hypertension** | 77.2 ± 9.5 | 84.7 ± 9.4 | 67.3 ± 8.8 | 167.3 ± 30.5 | 50.8 ± 10.9 | 30.7 ± 10.5 | 6541 ± 2625 | 3776 ± 1938 | 1684 ± 596 | 992 ± 369 | 0.61 ± 0.22 |

HR: heart rate; SDNN: standard deviation of all NN intervals; RMSSD: the square root of the mean of the sum of the squares of differences between adjacent NN intervals; pNN50: number of pairs of adjacent NN intervals differing by more than 50 ms divided by the total number of all NN intervals; TP: total power; VLF: very low-frequency power; LF: low-frequency power; HF: high-frequency power; HF/LF: Ratio HF to LF.

Unpaired t-test between healthy control and eight patient groups

aP-value < 0.05.

bP-value < 0.01.

cP-value < 0.001.

However, some patients met several group criteria that would be lost for further analysis. For risk calculation of an inappropriate sinus tachycardia, every single diagnosis of each patient was assigned to the 8 diagnosis groups. In summary we include 595 diagnosis of the 479 children.

## 24-h ECG and analysis of heart rate variability

Fundamental to every HRV data analysis is a group of successive normal RR intervals in sinus rhythm over a period of 24 h. In HVR these are called NN intervals to distinguish them from the RR intervals in cardiac arrhythmia. Measurement and interpretation of HRV parameters in the current sample were standardized according to the Task Force Guidelines [9]. Cardiac autonomic functioning was measured by 24-h Holter 12-bit digital ECG (Reynolds Pathfinder II, Spacelabs, Germany; 1024 scans/sec). Day- and nighttime periods were defined according to patient protocols. All Holter recordings were reviewed by the same experienced cardiologists (RB and CW) and were edited to validate the system's QRS labeling. Measures of HRV were calculated employing only NN intervals. QRS-complexes classified as noise were excluded from the data. A minimum of 23 h of analyzable data and minimum of 95% of analyzable NN intervals were required for data to be included. For time domain measures, mean NN interval, resulting heart rate, and HRV parameters outlined below were calculated.

For didactic reasons, in this study, we focused on the statistical analysis of the following four parameters:

1. **Heart rate:** The easiest, but very important, HRV parameter is the average sinus rhythm heart rate, since all other parameters are significantly affected by the heart rate.

2. **Standard deviation of all normal NN intervals in a time frame (SDNN):** This global HRV parameter represents the overall variability of the autonomic nervous system.

3. **Square root of the arithmetic mean of the squared deviation of successive normal NN intervals in a time frame (rMSSD):** This parameter is mainly influenced by the parasympathetic nervous system.

4. **Number of pairs of adjacent NN intervals differing by more than 50 ms, divided by the total number of all NN intervals, multiplied by 100 (pNN50):** This parameter is mainly influenced by the parasympathetic nervous system.

For frequency domain measures, beat-to-beat fluctuations were transformed to the frequency domain using Fast Fourier Transformation. Spectral power was determined over three frequency regions of interest: very low frequency (VLF, < 0.04 Hz), low frequency (LF, 0.04–0.15 Hz), and high frequency (HF, 0.15–0.4 Hz) with derived HF/LF ratios.

## Pharmacotherapy and nutritional supplements

Many patients received pharmacotherapy (e.g., psychostimulants, or growth hormones), others received nutritional intervention (e.g., nutritional refeeding in anorexia nervosa, or omega-3 fatty acid supplementation) with published effects on HRV [6]. In the current analysis we tested the effect of methylphenidate treatment on 24-h HRV in 19 children with a mean age of 11.4 ± 2.6 years (18 boys, 1 girl). The data was based on two Holter ECGs over a mean interval of 283 days after starting methylphenidate (N = 11) or ceasing an ongoing therapy (N = 8). All children received an extended-release formulation, 26% with a low dose immediate-release starter in the morning. The extended-release dosage was low (<0.5 mg/kg MPH) in 52.6%, intermediate (0.5 mg/kg to 0.9 mg/kg MPH) in 31.6%, and high (>0.9 mg/kg MPH) in 15.8% of the children.

After we realized that omega-3 fatty acid supplementation has an impact on reduced heart rate variability, 145 patients and/or parents decided for omega-3-fatty acid supplementation and a Holter ECG control 196 days later as an average in the last seven years. A group of 29 adolescents had sinus tachycardia > 95 bpm and was analyzed for this publication. The other 45 patients and or parents with sinus tachycardia > 95 bpm did not decide for omega-3-fatty acid supplementation or received a pharmacotherapy most of all with a beta blocker in children with hypertension or congenital heart disease. To exclude a selection bias between patients with inappropriate sinus tachycardia who got omega-3-fatty acid supplementation (N = 29) and those who received no supplements (N = 45), we compare all the parameters using a student t-test and found no significant difference (for example mean heart rate 99.4 ± 4.9 versus 99.8 ± 5.0 bpm), except a slightly lower diastolic blood pressure percentile in the omega-3-fatty acid supplementation group (46.6 ± 25.4% versus 59.9 ± 25.9%, p = 0,032).

If the supplementation was not covered by health insurance, patients purchased different products delivering 1–2 g fish oil per day from a retail store. The following dose recommendations were given: Children up to 8 years old should receive at least 400 mg Eicosapentaenoic acid (EPA) and Docosahexaenoic acid (DHA) as a suspension per day. Children who were able to swallow capsules should receive at least 800 mg EPA and DHA per day.

## Blood pressure

For blood pressure measurements we used oscillometric (automated) blood pressure measurement devices (Bionics Sentry™, South Korea), which have rapidly replaced sphygmomanometers in clinical practice. These devices are more ecologically friendly, easier to use, and eliminate potential sources of bias. During Bionics Sentry™ measurement, inflation is driven by a pumping system and deflation is driven by an electromagnetic control valve that allows rapid air release. Records are digitally displayed as systolic- and diastolic blood pressure, and heart rate. The internal bladder (inflatable area) of the cuff must encircle 90–100% of the circumference of the upper arm.

Norms for childhood blood pressure among normal-weight children (body mass index < 85th percentile, based on Centers for Disease Control and Prevention guidelines) as a function of age, sex, and height, were established using data from 49,967 children, included in the database of the National High Blood Pressure Education Program Working Group on High Blood Pressure in Children and Adolescents [10]. Blood pressure percentiles and the deviation from normal (z-score) values were calculated.

## Statistical analysis

Data were anonymized before accessing them for statistical analysis. Data were expressed as mean ± standard deviation. As most variables exhibited a normal distribution, between-group differences were assessed using parametric statistics. The study population was divided into eight diagnosis groups and one healthy control group. One-way analysis of variance was used to compare the differences between the healthy control group and the patient groups (Tables 1 and 2). We used a paired student t-test to analyze the effect of methylphenidate and omega-3-fatty acid supplementation on HRV (Tables 3 and 4). For risk stratification according to the different diagnosis groups we used a binary regression analysis (Table 5). Significant group differences were anticipated if the p-value was < 0.05. Data with significant bivariate correlations were used for linear regression analysis (Table 6 and S1 Table). All analyses were performed using IBM SPSS Statistics software, Version 25 (IBM Corp. IBM SPSS Statistics for Windows, Version 25.0, Armonk, NY, USA).

**Table 3. Methylphenidate (MPH) treatment and heart rate variability.**

| Patients (N = 19) | 24-h heart rate variability | | | Day time | | | Night time | | |
|---|---|---|---|---|---|---|---|---|---|
| | Baseline | MPH | p-value | Baseline | MPH | p-value | Baseline | MPH | p-value |
| **Mean heart rate [bpm]** | 91.3 ± 7.5 | 91.9 ± 8.0 | 0.771 | 96.0 ± 8.3 | 101.8 ± 10.8a | 0.02 | 86.7 ± 9.7 | 82.4 ± 9.0 | 0.081 |
| **RMSSD [ms]** | 32.5 ± 12.50 | 37.1 ± 17.8 | 0.053 | 29.8 ± 10.4 | 28.2 ± 12.1 | 0.287 | 35.1 ± 16.5 | 45.8 ± 24.3b | 0.004 |
| **pNN50 [%]** | 11.8 ± 8.4 | 14.1 ± 9.6 | 0.106 | 9.9 ± 6.4 | 8.7 ± 6.7 | 0.238 | 13.7 ± 11.8 | 19.4 ± 13.3a | 0.013 |
| **Total power [ms²]** | 3812 ± 2021 | 4246 ± 2396 | 0.344 | 3798 ± 2059 | 3857 ± 24367 | 0.887 | 3811 ± 2393 | 4640 ± 2545 | 0.125 |
| **VLF power [ms²]** | 2357 ± 1370 | 2534 ± 1836 | 0.631 | 2437 ± 1444 | 2425 ± 1957 | 0.974 | 2264 ± 1692 | 2653 ± 1852 | 0.369 |
| **LF power [ms²]** | 981 ± 608 | 1165 ± 611 | 0.060 | 955 ± 570 | 1050 ± 571 | 0.215 | 1005 ± 677 | 1278 ± 723a | 0.04 |
| **HF power [ms²]** | 416 ± 223 | 490 ± 290 | 0.195 | 348 ± 211 | 330 ± 202 | 0.607 | 483 ± 302 | 647 ± 431 | 0.069 |
| **HF/LF Ratio** | 0.48 ± 0.27 | 0.62 ± 0.35 | 0.968 | 0.42 ± 0.26 | 0.41 ± 0.21 | 0.521 | 0.81 ± 0.34 | 0.83 ± 0.38 | 0.853 |

SDNN: standard deviation of all NN intervals; RMSSD: the square root of the mean of the sum of the squares of differences between adjacent NN intervals; pNN50: number of pairs of adjacent NN intervals differing by more than 50 ms divided by the total number of all NN intervals; TP: total power; VLF: very low-frequency power; LF: low-frequency power; HF: high-frequency power; MPH: methylphenidate treatment.

Paired t-test between baseline and methylphenidate treatment

[a]P-value < 0.05.

[b]P-value < 0.01.

## Results

The anthropometric data are displayed in Table 1. Patients in the nine groups were all of comparable age. By definition the healthy control group patients had normal height, weight, body mass index, systolic and diastolic blood pressures. According to the group definitions mean height, weight, and body mass index were significantly different in the short stature, obesity, anorexia nervosa, and constitutional thinness groups. Systolic/diastolic blood pressures were hypertensive in the hypertension (≥95th percentile) and pre-hypertension (≥90th percentile) groups. In addition to these expected results, the following significant differences compared to the healthy control group were noticeable in the patient groups:

**Table 4. Effect of Omega-3-fatty acid supplementation on heart rate variability in 29 adolescents with inappropriate sinus tachycardia.**

| Patients (N = 29) | 24-h heart rate variability | | | Day time | | | Night time | | |
|---|---|---|---|---|---|---|---|---|---|
| | Baseline | Omega-3-FA | p-value | Baseline | Omega-3-FA | p-value | Baseline | Omega-3-FA | p-value |
| **Mean heart rate [bpm]** | 99.4 ± 4.9 | 90.1 ± 7.7c | <0.001 | 105.0 ± 7.2 | 98.4 ± 7.7c | 0.003 | 87.5 ± 9.9 | 75.8 ± 6.0c | <0.001 |
| **RMSSD [ms]** | 22.6 ± 7.9 | 31.8 ± 12.6c | <0.001 | 21.6 ± 10.5 | 27.1 ± 12.0a | 0.018 | 34.4 ± 24.0 | 47.1 ± 26.4a | 0.011 |
| **pNN50 [%]** | 7.7 ± 6.4 | 13.3 ± 10.7b | 0.003 | 5.3 ± 5.1 | 8.8 ± 7.3a | 0.02 | 11.7 ± 13.0 | 22.1 ± 17.8b | 0.002 |
| **Total power [ms²]** | 2587 ± 1641 | 3522 ± 2206b | 0.006 | 2292 ± 2027 | 3259 ± 2344b | 0.002 | 3227 ± 2852 | 4435 ± 3323a | 0.043 |
| **VLF power [ms²]** | 1337 ± 727 | 1870 ± 1348a | 0.014 | 1115 ± 846 | 1758 ± 1679b | 0.007 | 1753 ± 1536 | 2274 ± 1913 | 0.117 |
| **LF power [ms²]** | 689 ± 474 | 919 ± 586b | 0.008 | 635 ± 401 | 856 ± 482b | 0.003 | 851 ± 1001 | 1115 ± 1001 | 0.054 |
| **HF power [ms²]** | 435 ± 561 | 599 ± 613a | 0.01 | 374 ± 794 | 470 ± 723a | 0.045 | 552 ± 651 | 900 ± 676a | 0.027 |
| **HF/LF Ratio** | 0.56 ± 0.39 | 0.59 ± 0.36 | 0.559 | 0.41 ± 0.42 | 0.46 ± 0.39 | 0.376 | 0.68 ± 0.37 | 0.81 ± 0.45 | 0.164 |

SDNN: standard deviation of all NN intervals; RMSSD: the square root of the mean of the sum of the squares of differences between adjacent NN intervals; pNN50: number of pairs of adjacent NN intervals differing by more than 50 ms divided by the total number of all NN intervals; TP: total power; VLF: very low-frequency power; LF: low-frequency power; HF: high-frequency power; Omega-3-FA: omega-3-fatty acid supplementation.

Paired t-test between baseline and omega-3-Fatty Acid Supplementation

[a]P-value < 0.05.

[b]P-value < 0.01.

[c]P-value < 0.001.

**Table 5. Binary regression analysis.**

| | Regression coefficentB | Standard Error | Wald | Significance | Exp (B) | 95% Confidence interval EXP(B) | |
|---|---|---|---|---|---|---|---|
| | | | | | | Lower limit | Upper limit |
| Sex (male = 1) | -0,524 | 0,294 | 3,186 | 0,074 | 0,592 | 0,333 | 1,053 |
| Congenital Heart Disease | 0,306 | 0,461 | 0,441 | 0,507 | 1,358 | 0,55 | 3,353 |
| Attention Deficit Disorder | 1,265 | 0,335 | 14,297 | 0 | 3,543 | 1,839 | 6,826 |
| Anorexia Nervosa | -1,191 | 1,051 | 1,285 | 0,257 | 0,304 | 0,039 | 2,382 |
| Short Stature | 0,637 | 0,485 | 1,722 | 0,189 | 1,891 | 0,73 | 4,897 |
| Const. Thinness | -0,11 | 0,669 | 0,027 | 0,87 | 0,896 | 0,241 | 3,325 |
| Obesity | 0,359 | 0,312 | 1,322 | 0,25 | 1,431 | 0,777 | 2,638 |
| Hypertension | 0,724 | 0,311 | 5,431 | 0,02 | 2,062 | 1,122 | 3,79 |
| Pre Hypertension | 0,918 | 0,453 | 4,099 | 0,043 | 2,503 | 1,03 | 6,084 |

1) Obese children and children with attention deficit disorder had significantly hypertensive systolic blood pressures on average.

2) Children with anorexia nervosa had significantly reduced systolic blood pressures on average.

The HRV data are displayed in Table 2. Children with attention deficit disorder had significantly elevated heart rates on average by day. Obese children had significantly elevated heart rates at night. Children with anorexia nervosa had significantly lower heart rates both day and night. Higher heart rates in obese adolescents were associated with a lower global HRV SDNN.

**Table 6. Linear regression analysis: Impact of age, height, body mass index, and heart rate variability on the systolic blood pressure.**

| Dependent variable: systolic blood pressure percentile | | Unstandardized coefficients | | Standardized coefficients | T | Sig. |
|---|---|---|---|---|---|---|
| | | B | Std. error | Beta | | |
| Time domain analysis: $R^2 = 0.265$ | (Constant) | 101.293 | 10.533 | | 9.616 | 0.000 |
| | Height SDS | −0.123 | 0.161 | −0.032 | −0.761 | 0.447 |
| | BMI SDS | 6.465 | 0.745 | 0.385 | 8.678 | 0.000 |
| | Age | −0.173 | 0.637 | −0.012 | −0.272 | 0.786 |
| | NN 24h | −0.053 | 0.015 | −0.240 | −3.614 | 0.000 |
| | rMSSDs 24h | −0.285 | 0.129 | −0.218 | −2.208 | 0.028 |
| | SDNN 24h | 0.337 | 0.115 | 0.277 | 2.931 | 0.004 |
| Frequency domain analysis: $R^2 = 0.256$ | (Constant) | 109.261 | 11.024 | | 9.911 | 0.000 |
| | Height SDS | −0.096 | 0.163 | −0.025 | −0.591 | 0.555 |
| | BMI SDS | 6.470 | 0.753 | 0.385 | 8.588 | 0.000 |
| | Age | 0.000 | 0.635 | 0.000 | 0.001 | 0.999 |
| | NN 24h | −0.059 | 0.014 | −0.268 | −4.306 | 0.000 |
| | TP 24h | 0.002 | 0.007 | 0.204 | 0.237 | 0.813 |
| | VLF 24h | −0.001 | 0.007 | −0.110 | −0.180 | 0.857 |
| | LF 24h | 0.002 | 0.008 | 0.046 | 0.236 | 0.814 |
| | HF 24h | −0.003 | 0.008 | −0.087 | −0.386 | 0.699 |
| | HF/LF 24h | 0.392 | 0.619 | 0.027 | 0.633 | 0.527 |

Lower heart rates in adolescents with anorexia nervosa were associated with higher global HRV SDNN.

Significantly lower pNN50 and RMSSD values in adolescents with congenital heart defects, attention deficit disorder, and obesity may indicate a lower vagus activity, whereas significantly higher pNN50 and RMSSD values in adolescents with anorexia nervosa may indicate a higher vagus activity. However, these interpretations were not supported by high-frequency power data that were only significantly reduced in children with attention deficit disorder. Changes in high-frequency power were accompanied by concordant changes in low-frequency power, whereas the HF/LF ratio remained within the same range in all groups.

For an age-specific definition of an inappropriate mean 24-h sinus heart rate in adolescents we used the data of the healthy control group who had a mean heart rate of 78.2 bpm with a double standard deviation of 16.8 bpm.

In summary, 74 adolescents had a mean 24-h heart rate $\geq$ 95 bpm, our cut-off for IST in adolescents. This cut-off is 5 bpm higher than that for adults with inappropriate sinus tachycardia. A logistic regression was performed to ascertain the effects of gender, congenital heart disease, attention deficit disorder, anorexia nervosa, short stature, constitutional thinness, obesity, hypertension, pre-hypertension on the likelihood that participants have IST. The model explained 11.0% (Nagelkerke $R^2$) of the variance in IST and correctly classified 86.3% of cases. As shown in Table 5, we found the risk of IST to be significantly high in adolescents with attention deficit disorder (OR = 3.5, p<0.001), pre-hypertension (OR = 2.5, p = 0.043) and hypertension (OR = 2.1, p = 0.02); insignificantly enhanced in children with short stature (OR = 1.9, p = 0.19), surgically-treated congenital heart disease (OR = 1.4, p = 0.51) and obesity without hypertension (OR = 1.4; p = 0.25); and negligible in adolescents with anorexia nervosa (OR = 0.3, p = 0.26) and constitutional thinness (OR = 0.9, p = 0.89).

Of the 22 children with IST and attention deficit disorder, 13 formed part of the group of 69 children treated with methylphenidate, 6 formed part of the group of 7 children treated with atomoxetine, and 2 formed part of the group of 3 children treated with amphetamine.

These data suggest that IST in adolescents with attention deficit disorder is related to stimulant treatment, with the strongest association being to atomoxetine and amphetamine treatment. However, when we analyzed the effect of methylphenidate treatment on 24-h HRV in 19 adolescents, using data based on two Holter ECGs measured across a mean interval of 283 days after starting methylphenidate (N = 11) or stopping an ongoing therapy (N = 8), the mean 24-h heart rate of 91.3 ± 7.5 bpm remained unchanged with methylphenidate treatment at 91.9 ± 8.0 bpm (Table 3). This surprising result indicates that the well-known heart rate increase after the daytime administration of methylphenidate from 96.0 ± 8.3 to 101.8 ± 10.8 bpm (p = 0.020) was completely compensated for by an equal heart rate decrease at night from 86.7 ± 9.7 to 82.4 ± 9.0 bpm. This effect on the circadian heart rate pattern is illustrated in Fig 1.

We tested the impact of anthropometric measurements and HRV on the systolic blood pressure percentile using a linear regression analysis. The body mass index standard SDS value and the mean 24-h NN interval (heart rate = 60000/NN interval) were used to determine the systolic blood pressure percentile (Table 6). The inclusion of the vagus parameter, RMSSD, and the global HRV parameter, SDNN, slightly improved this model, but not the frequency domain analysis. The impact of the heart rate (or NN Interval) on systolic blood pressure was the same for daytime or nighttime heart rates.

## Discussion

IST is a common cardiovascular risk factor in adolescence–74 of 479 (15.4%) adolescents in the current study had an elevated 24-h heart rate $\geq$ 95 bpm. At first, this result appeared to be

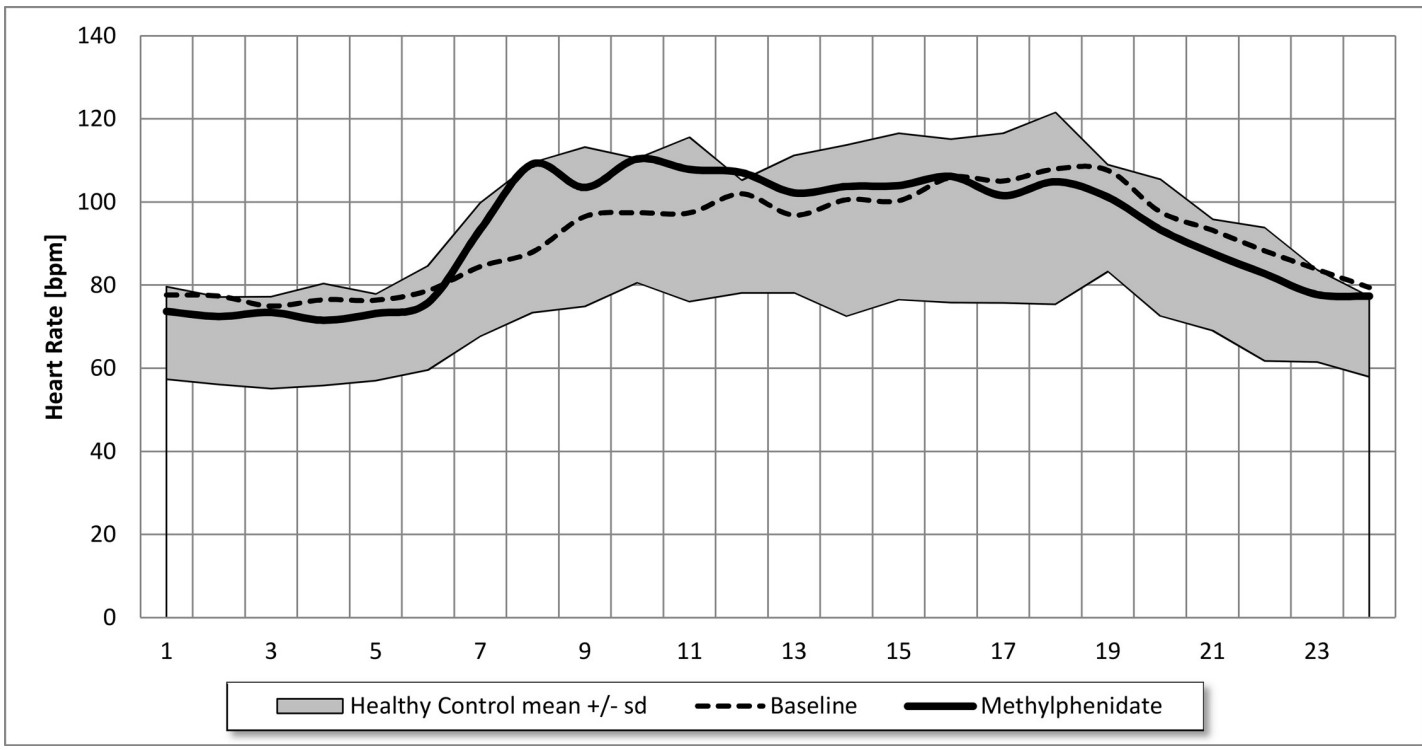

**Fig 1. Effects of Methylphenidate treatment on circadian heart rate in 19 adolescents.** We prescribed omega-3 fatty acid supplementation and patients complied in 29 of the 74 adolescents with inappropriate sinus tachycardia, as shown in Table 4, and we found highly significant effects on heart rate both by day and at night. The significant heart rate decrease of 8.3 bpm over 24 h, 6.6 bpm in daytime, and 11.7 bpm at night, related to a significantly higher HRV. Significantly higher RMSSD, pNN50, and high-frequency power values indicate a higher vagus activity after omega-3-fatty acid supplementation. However, as shown in other studies [6], omega-3-fatty acid supplementation improves HRV across the whole power spectrum of the Fast Fourier Analysis with nearly no effect on the HF/LF ratio. This could be an evidence of a specific effect of omega-3 fatty acid supplementation in children with autonomic dysfunction that not only depends on the autonomic system but also upon its effect on specific ion channels of the sinoatrial node.

related to the high use of psychostimulants (N = 79) in adolescents with attention deficit disorder. However, our data indicate that methylphenidate did not increase 24-h heart rates in 19 of 69 adolescents who had a Holter ECG with and without methylphenidate. This surprising result indicates that the well-known heart rate increase after daytime administration of methylphenidate from 96.0 ± 8.3 to 101.8 ± 10.8 bpm (p = 0.02) is completely compensated for by an equal heart rate decrease at night from 86.7 ± 9.7 to 82.4 ± 9.0 bpm, probably due to vagus activation with significantly higher RMSSD and pNN50 values.

Elevated heart rates, at least in adolescents with attention deficit disorder, seems to be a result of a genuine autonomic dysfunction with significantly reduced HRV. The risk of an IST is insignificant in children with surgically-treated congenital heart disease, short stature and obesity without arterial hypertension, and is negligible in adolescents with anorexia nervosa and constitutional thinness (Table 5).

We found significantly reduced heart rates and blood pressures in adolescents with anorexia nervosa, but not in adolescents with constitutional thinness, despite the same average body mass index (Tables 1 and 2). As we anticipated, caloric intake had an effect on both HRV and blood pressure, with low caloric intake inducing low heart rates and hypotension in individuals with anorexia nervosa and high caloric intake inducing high heart rates and hypertension in obese adolescents. We further found that children with constitutional thinness had normal caloric intake and normal heart rates and blood pressures (Tables 1 and 2). Calorie

reduction is probably the most effective treatment for reducing elevated heart rates and blood pressures in obese patients.

Linear regression analysis in the full cohort of 479 adolescents, clearly showed that the systolic blood pressure percentile depends primarily on the body mass index and mean 24-h heart rate. This result is in good accordance with the data from adults with IST who had a high risk of arterial hypertension. In summary, the more established cardiovascular risk factor, arterial blood pressure, and elevated mean heart rate probably predict cardiovascular risk. Our observations concur with those of the Swedish register data which showed that an elevated resting heart rate in adolescence is an important cardiovascular risk factor for later all-cause mortality primarily due to the higher risk of heart failure. Resting heart rate is independently associated with longevity, even when familial factors are controlled, such as in twin studies [11].

We investigated potential treatments for adolescents with IST with heart rates $\geq$ 95 bpm. We stopped psychostimulants in some children with attention deficit disorder, but this seems to be effective only in adolescents treated with atomoxetine and amphetamine and not in adolescents treated with methylphenidate. Much more effective was the omega-3-fatty acid supplementation of 29 adolescents with inappropriate sinus tachycardia, who showed a significant heart rate decrease by 8.3 bpm over 24 h, 6.6 bpm at daytime, and 11.7 bpm at night, together with significantly higher HRV. We did not systematically measure blood pressures after omega-3-fatty acid supplementation, but the effect on hypertension seems to be lower than that on heart rate.

After exclusion of hypertension in obese children, the remaining 53 adolescents with arterial hypertension had normal body mass indices and heart rate variabilities. Most of these adolescents received pharmacotherapy to treat hypertension. Patients with pre-hypertension had normal HRVs on average and probably no longer needed pharmacotherapy but only 24-h blood pressure monitoring.

Adolescents with surgically-treated congenital heart disease had nearly normal HRVs probably due to the close monitoring and pharmacotherapy with beta blockers, angiotensin-converting enzyme inhibitors, and angiotensin receptor antagonists.

Children with short stature due to small gestational age, syndromes, heart defects, and growth hormone deficiency, but not due to constitutional growth delay, had a reduced HRV as recently published [3]. There is no additional information in the current analysis.

We are aware that there are many open questions regarding the pathophysiology of IST as discussed in the literature [12]. We currently doubt our own hypothesis that the changes of HRV in children with IST are an expression of an autonomic imbalance between sympathicus and vagus. These very sophisticated considerations are not included in this publication written for clinicians. Instead we refer the reader to our current open access publication [13]. The question is if the reduced HRV in adolescents with IST across the whole power spectrum of the Fast Fourier Analysis with nearly no changes of the HF/LF ratio, not only depends on the autonomic system, but upon changes of the ion channels of the sinoatrial node. This important pathophysiological consideration may explain why omega-3 fatty acids as well as unusual channel blockers, like ivabradine, are even more effective in the treatment of IST than beta blockers.

## Study limitations

We divided this large patient group into many (not mutually disclosing) groups according diagnosis, height, body mass index and blood pressure. This methodological problem was part of the reviewing process. We included a binary logistic regression analysis which considers all

diagnoses of the patients we examined and gives us the opportunity to declare odds ratios for the risk of IST.

This is a retrospective analysis including Holter ECG's from 2005. During these 15 years, we realized the impact of nutrition and omega-3-fatty acid supplementation on heart rate variability. The multiple individual therapeutic decisions cannot be a part of this publication. We focus on methylphenidate therapy in ADHD and omega-3-fatty acid supplementation in autonomic dysfunction. Due to the authors'specialisation in pediatric cardiology, nearly all children had in part multiple ECG and echocardiographies that are not analyzed for this publication. The reader may be certain that none of the patients have an unrecognized heart disease.

We collected the raw data and we were able to transfer new methodological knowledge to older data sets. Today, we emphasize the importance of the mean heart rate, especially when it comes to risk assessment of cardiovascular endpoints. Heart rate significantly influences HRV due to both physiological and mathematical reasons. Moreover, physicians understand the heart rate much better than the sometimes-complicated method of HRV analysis. We try to sensitize pediatricians to the problem of heart rate regulation like inappropriate sinus tachycardia and offer an easily understandable cut off 95 bpm in adolescence.

However, the retrospective nature of the study–especially regarding omega-3-fatty acid supplementation–is a strong risk factor for "regression to the mean" as confounding mechanism. Moreover, a selection bias in children who got omega-3-fatty acid supplementation lies on the fact that their parents self-select themselves for the treatment, prove a higher motivation for the therapy and remain susceptible to a placebo effect. This means that the results of our analysis are hypothesis generating and deserve further exploration in a randomized control study.

## Supporting information

**S1 Table.**
(DOCX)

**S2 Table.**
(DOCX)

## Acknowledgments

We thank the ªBlaschek foundationº who supports us by covering the publication fees for this research project. Grant Recipient is Dr. Reiner Buchhorn. The funders had no role in study design, data collection and analysis, decision to publish, or preparation of the manuscript.

## Author Contributions

**Conceptualization:** Reiner Buchhorn, Semanur Gündogdu, Christian Willaschek.

**Data curation:** Reiner Buchhorn, Christoph Baumann, Semanur Gündogdu, Ulla Rakowski.

**Formal analysis:** Reiner Buchhorn, Christoph Baumann, Semanur Gündogdu, Ulla Rakowski.

**Funding acquisition:** Reiner Buchhorn.

**Investigation:** Reiner Buchhorn, Christoph Baumann, Semanur Gündogdu, Christian Willaschek.

**Methodology:** Reiner Buchhorn, Semanur Gündogdu, Christian Willaschek.

**Project administration:** Reiner Buchhorn.

**Resources:** Reiner Buchhorn, Semanur Gündogdu, Christian Willaschek.

**Software:** Reiner Buchhorn, Christoph Baumann, Semanur Gündogdu, Ulla Rakowski.

**Supervision:** Reiner Buchhorn.

**Validation:** Reiner Buchhorn, Semanur Gündogdu, Christian Willaschek.

**Visualization:** Reiner Buchhorn, Semanur Gündogdu.

**Writing – original draft:** Reiner Buchhorn, Semanur Gündogdu.

**Writing – review & editing:** Reiner Buchhorn, Christian Willaschek.

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
