## [Decision Letter · Decision Letter 0]

20 May 2020

PONE-D-20-09485

Diagnosis and management of an inappropriate sinus tachycardia in adolescence based upon a Holter ECG: A retrospective analysis of 479 patients

PLOS ONE

Dear Dr. Willaschek,

Thank you for submitting your manuscript to PLOS ONE. After careful consideration, we feel that it has merit but does not fully meet PLOS ONE’s publication criteria as it currently stands. Therefore, we invite you to submit a revised version of the manuscript that addresses the points raised during the review process.

Specifically, clinical information and stastistical methods should be implemented.

We would appreciate receiving your revised manuscript by Jul 04 2020 11:59PM. To enhance the reproducibility of your results, we recommend that if applicable you deposit your laboratory protocols in protocols.io, where a protocol can be assigned its own identifier (DOI) such that it can be cited independently in the future. For instructions see: http://journals.plos.org/plosone/s/submission-guidelines#loc-laboratory-protocols

We look forward to receiving your revised manuscript.

Kind regards,

Antonio Cannatà

Academic Editor

PLOS ONE

Journal Requirements:

1. Please ensure that your manuscript meets PLOS ONE's style requirements, including those for file naming. The PLOS ONE style templates can be found at https://journals.plos.org/plosone/s/file?id=wjVg/PLOSOne_formatting_sample_main_body.pdf andhttps://journals.plos.org/plosone/s/file?id=ba62/PLOSOne_formatting_sample_title_authors_affiliations.pdf

Additional Editor Comments (if provided):

Reviewers' comments:

Reviewer's Responses to Questions

**Comments to the Author**

1. Is the manuscript technically sound, and do the data support the conclusions?

Reviewer #1: Partly

Reviewer #2: Partly

5. Review Comments to the Author

Reviewer #1: General comments:

This is a retrospective study evaluating 479 children for the association of inappropriate sinus tachycardia (IST, n=79) with heart rate variability (HRV) in Holter-ECG. Healthy controls (n=66) are compared to 8 different groups with attention deficit disorder (ADD, n=86), obesity (n=130), short stature (n=15), congenital heart disease (n=55), anorexia nervosa (n=34), constitutional thinness (n=20), hypertension (n=53) or pre-hypertension (n=13). The IST risk was high in those with ADD, obesity, and short stature and low in those with corrected congenital heart disease and hypertension. IST was associated with a decreased global HRV and hypertension. Omega-3-fatty acid supplementation was associated with decreased HR and increased HRV.

The authors are commended for their elaborate work and data analysis. This is an interesting work, but some issues need to be addressed prior to considering publication in the PLOS ONE Journal.

Specific comments:

1. Methods: Dividing patients into many (not mutually disclosing) groups and separating those with low height (short stature), (constitutional) low or high BMI (obese) and high blood pressure (BP, pre-/hypertension) creates some methodological problems, since further comparisons do not adjust for the differences for such factors. As the authors point out those with obesity and ADD had higher BP, and those with anorexia nervosa had lower BP. Similarly it seems that high BMI/obesity was also present to some extent in other groups too. Please consider using logistic regression to calculate the adjusted risk for IST and report OR and confidence intervals along with p values.

2. Methods: How were the 66 healthy controls chosen? What was the indication for Holter-ECG in these individuals? Consider possible implications/selection bias.

3. Results/Table 1: Please report what is the percentage of obese or hypertensive children in each group and how many children in each group had IST.

4. Methods/Results/Figure 1: Please report how the IST risk was calculated. It seems like the depicted risk represents the percentage/prevalence of IST cases in each group (e.g. 22/86=26% for ADD). However, as described in point 1 this is unadjusted for BP/BMI/height and could be misleading. For example since those with obesity and ADD had high BP, we do not know to what extent BP could bias these findings.

5. Results/Limitations: Omega-3-fatty acid supplementation was up to the patient`s choice. Therefore a selection bias could confound the effect of supplementation. Additionally, since IST has been also associated with psychosocial distress, a strong placebo effect could partially account for the same effect. Please compare baseline characteristics of those who choose supplementation vs. those who did not and comment accordingly in the limitations section.

6. Results/Table 3 & 4: Please report on the sympathetic activity (LF/HF ratio) too. Comment if significant differences become apparent

7. Results/Table 4: Please report the p values (some cells have missing/zero values)

8. Methods/Limitations: HR significantly influences HRV due to both physiological and mathematical reasons. Thus changes in HR should be taken into account when evaluating HRV changes (e.g. obese vs. anorectic children or ADD on/off MPH). This could be implemented in a sensitivity analysis or acknowledged in the limitations section.

Reviewer #2: The authors of this manuscript analysed the prevalence of inappropriate sinus tachycardia (IST) in adolescence.

Despite the topic is interesting, the are some critical point:

1) The diagnosis of IST seems to be based only on a single 24-h Holter 12-bit digital ECG. More measurements would have been more appropriate for a better diagnosis. Please report it in the limitation of the study.

2) Any information about the clinic follow-up of these patients has been reported. If it is possible, follow-up data should be included.

3) ECG, echocardiographic or anamnestic information is not be included in the analysis. This is an important limitation for this study.

4) The tables should be improved and integrated with for clinical information of the patients.

---

## [Author Response · Author response to Decision Letter 0]

12 Jul 2020

Answers to Reviewer 1

 1. Methods: Indeed, dividing our patients into many groups with a broad overlap leads to important methodological problems. We performed a binary logistic regression analysis as proposed by reviewer 1. Heavy work - but a significant improvement of our publication. We deleted Figure 1 due to the misleading interpretation based on IST percentages in the diagnosis groups. We included a new table 5 with the results of the binary logistic regression analysis. The new results according to the Odds ratios are declared in the abstract:

“We found the risk of IST to be high in adolescents with attention deficit disorder (OR=3.5,p<0.001), children with short stature (OR=1.9,p=0.19) and hypertension (OR=2.1,p=0.02); lower in surgically-treated congenital heart disease (OR=1.4,p=0.51) and obesity without hypertension (OR=1.4;p=0.25); and negligible in adolescents with anorexia nervosa (OR=0.3, p=0.26) and constitutional thinness (OR=0.9,p=0.89).

 2. ” We explain the collection of the healthy control group and the citation of the published data and the ethic approvement (Reference 8).

 3. Results: We include a new table with the number and percentage of each diagnosis as supplement to our publication (Tablediagnosisgroups). 

 4. See Point 1

 5. In a new statistic we compare adolescents who received Omega-3-FA with those who received no Omega-3-FA and found no significant group difference. Supplement to our publication (TableOmega3).

 6. The HF/LF ratios are calculated and included in table 3&4.

 7. The p-values are given as <0.001 instead of a zero

 8. See study limitations

Reviewer 2

Reviewer #2: The authors of this manuscript analysed the prevalence of inappropriate sinus tachycardia (IST) in adolescence.

Despite the topic is interesting, the are some critical point:

1) The diagnosis of IST seems to be based only on a single 24-h Holter 12-bit digital ECG. More measurements would have been more appropriate for a better diagnosis. Please report it in the limitation of the study.

2) Any information about the clinic follow-up of these patients has been reported. If it is possible, follow-up data should be included.

3) ECG, echocardiographic or anamnestic information is not be included in the analysis. This is an important limitation for this study.

4) The tables should be improved and integrated with for clinical information of the patients.

Answers to reviewer 2

 1. IST depends on definition based upon the mean 24 hour heart rate >90 bpm or daily heart rates >100 bpm in adults. We adapt this definition to adolescents based upon our normal values. This definition was proofed in each patient.

 2. We focused the follow up on a second Holter ECG according to pharmacologic or nutritional intervention with methylphenidate therapy or omega-3-fatty acid supplementation. That’s a lot for the first big study to inappropriate sinus tachycardia in childhood. Some authors frighten that the current paper is just overwhelmed. Calculating further follow up data would weaken the statistical power.

 3. Nearly all patients had ECG and echocardiographic data to exclude or proof heart defects or cardiomyopathies (see limitations).

 4. We include a supplement (Tablediagnosisgroups) that declare the individual diagnosis and percentages of IST, Obesity, Short Stature and hypertension. The Editor should decide weather to include this data as a 7th table in the original publication.

---

## [Decision Letter · Decision Letter 1]

31 Jul 2020

PONE-D-20-09485R1

Diagnosis and management of an inappropriate sinus tachycardia in adolescence based upon a Holter ECG: A retrospective analysis of 479 patients

PLOS ONE

Dear Dr. Willaschek,

Thank you for submitting your manuscript to PLOS ONE. After careful consideration, we feel that it has merit but does not fully meet PLOS ONE’s publication criteria as it currently stands. Therefore, we invite you to submit a revised version of the manuscript that addresses the points raised during the review process.

We look forward to receiving your revised manuscript.

Kind regards,

Antonio Cannatà

Academic Editor

PLOS ONE

Reviewers' comments:

Reviewer's Responses to Questions

**Comments to the Author**

1. If the authors have adequately addressed your comments raised in a previous round of review and you feel that this manuscript is now acceptable for publication, you may indicate that here to bypass the “Comments to the Author” section, enter your conflict of interest statement in the “Confidential to Editor” section, and submit your "Accept" recommendation.

Reviewer #1: (No Response)

Reviewer #2: All comments have been addressed

2. Is the manuscript technically sound, and do the data support the conclusions?

Reviewer #1: Partly

Reviewer #2: Yes

3. Has the statistical analysis been performed appropriately and rigorously? 

Reviewer #1: Yes

Reviewer #2: Yes

4. Have the authors made all data underlying the findings in their manuscript fully available?

Reviewer #1: Yes

Reviewer #2: Yes

5. Is the manuscript presented in an intelligible fashion and written in standard English?

Reviewer #1: Yes

Reviewer #2: Yes

6. Review Comments to the Author

Reviewer #1: General comments:

This is a revised manuscript about a retrospective study evaluating 479 children for the association of inappropriate sinus tachycardia (IST, n=79) with heart rate variability (HRV) in Holter-ECG. Healthy controls (n=66) are compared to 8 different groups with attention deficit disorder (ADD, n=86), obesity (n=130), short stature (n=15), congenital heart disease (n=55), anorexia nervosa (n=34), constitutional thinness (n=20), hypertension (n=53) or pre-hypertension (n=13). IST risk was higher in adolescents with attention deficit disorder (OR=3.5,p<0.001), hypertension (OR=2.1,p=0.02) and pre-hypertension, but did not reach statistical significance in other groups. IST was associated with decreased HRV and omega-3-fatty acid supplementation was associated with decreased HR and increased HRV in adolescents with IST.

The authors are commended for sharing their data and for their efforts on the revision of the manuscript. The statistical methods have significantly improved. The interpretation and discussion of the results though should appropriately address some limitations:

Specific comments:

1. Methods: Thank you for using logistic regression. Please notice that for the groups that the effect did not reach a statistical significance there is not enough evidence for an association. As such it is not correct to claim that IST risk was slightly increased in children with short stature 32 (OR=1.9,p=0.19), surgically-treated congenital heart disease (OR=1.4,p=0.51) and obesity 33 without hypertension (OR=1.4;p=0.25). The confidence interval of the ORs reveals the spread of that probability and why the p value is not significant. Interestingly, pre-hypertension (OR 2.5, p=0.04) reached significance but was not commented. Please correct accordingly the text/claims in the results and discussion sections.

2. Methods: The authors report that patients with IST who got omega-3-fatty acid supplementation (N=29) and those who received no supplements (N=45) did not differ in their baseline parameters, but the table “Omega3” shows higher systolic blood pressure (77±26 vs. 76±30 mmHg, p=0.03) for the supplement group. This is interesting, especially since the no-supplement groups got mostly beta-blockers and had hypertension or congenital heart disease. Thus, hypertension could bias the results of the omega-3 supplementation. Did all 29+45 patients have a follow-up Holter? All after 196 days? Please comment (and correct the structure of the table; every comparison has a P value, what is the 2nd P value?)

3. Discussion: The authors report that “elevated heart rates, at least in adolescents with attention deficit disorder, is a result of a genuine autonomic dysfunction with significantly reduced HRV”. Before one assumes causality has to acknowledge the alternative explanations. Could Methylphenidate (stimulant medication) be the reason for the increased heart rate that in term leads to a reduced HRV? Please comment appropriately in the Discussion/Limitations section.

4. Limitations: The selection bias in those who got omega-3-fatty acid supplementation lies on the fact that they self-select themselves for the treatment, prove a higher motivation for changes/therapy and remain susceptible to a placebo effect. This is a major limitation that should be mentioned in text for the readers along with weather their medication (Methylphenidate) changed along with the omega-3 supplements.

5. Limitations: The retrospective nature of the study, especially in regard to omega-3 supplementation with IST (where increased heart rate is a selection criterion), introduces a strong risk for “regression to the mean” as confounding mechanism. This means that the results of this analysis are hypothesis generating and deserve further exploration in a randomized control study.

Reviewer #2: The manuscript seems to be improved. It would be interesting following these young-patients during the growth.

7. PLOS authors have the option to publish the peer review history of their article (what does this mean?). If published, this will include your full peer review and any attached files.

Reviewer #1: No

Reviewer #2: No

---

## [Author Response · Author response to Decision Letter 1]

2 Aug 2020

Dear Editor,

Thank you again for the careful review of our manuscript.

The specific comments of Reviewer#1 are included in our revision:

1.

Indeed, the Odds ratios are not significantly elevated in the kids with short stature, congenital heart disease and obesity without hypertension – we correct the text. Hypertension and prehypertension are significantly more common in kids with inappropriate sinus tachycardia. This fact is in good accordance with data from adults (1) and underline that inappropriate sinus tachycardia is an important cardiovascular risk factor.

2.

 We proof this table and the statistics that is part of the supplement and not the original publication. There is only a slight difference in the diastolic blood pressure percentile. Furthermore, we demonstrate the absolute blood pressure values and there is only a difference of 2.9 mmHg systolic blood pressure and 5.2 mmHg diastolic blood pressure. I declare this difference in the publication, but I cannot imagine that this is a real bias. I improve this table and delete the two tailed p-values. 

Children who didn’t receive omega-3-fatty acids had no follow-up Holter ECG, except those who need a pharmacotherapy.

3.

The question if methylphenidate treatment is the cause of inappropriate sinus tachycardia in children with attention deficit disorder is very important and our first hypothesis. That is the reason why we test the kids with and without methylphenidate in a second Holter ECG. To our big surprise that is not the case as shown in table 3 (heart rate : 91.3 ± 7.5 bpm versus 91.9 ± 8.0 bpm). This is a very good information for millions of children who are treated with methylphenidate. To the best of our knowledge, we are the first who proof this hypothesis as everybody thinks that methylphenidate increases the heart rate as shown in the studies with single measurement after methylphenidate intake.

4 + 5.

We change the paragraph study Limitations:

Study Limitation

We must divide this large patient group into many (not mutually disclosing) groups according diagnosis, height, body mass index and blood pressure. This methodological problem was part of the reviewing process. We include a binary logistic regression analysis which considers all diagnoses of the patients we examined and give us the opportunity to declare odds ratios for the risk of IST.

This is a retrospective analysis including Holter ECG’s from 2005. During these 15 years, we realize the impact of nutrition and omega-3-fatty acid supplementation on heart rate variability. The multiple individual therapeutic decisions cannot be a part of this publication. We focus on methylphenidate therapy in ADHD and omega-3-fatty acid supplementation in autonomic dysfunction. As pediatric cardiologists, nearly all children had in part multiple ECG and echocardiographies that are not analyzed for this publication. The reader may be certain that none of the patients have an unrecognized heart disease. 

We collect the raw data and we were able to transfer new methodological knowledge to older data sets. Today, we emphasize the importance of the mean heart rate, especially when it comes to risk assessment of cardiovascular endpoints. Heart rate significantly influences HRV due to both physiological and mathematical reasons. Moreover, physicians understand the heart rate much better than the sometimes-complicated method of HRV analysis. We try to sensitize pediatricians to the problem of heart rate regulation like inappropriate sinus tachycardia and offer an easily understandable cut off 95 bpm in adolescence.

However, the retrospective nature of the study – especially regarding omega-3-fatty acid supplementation – is a strong risk factor for “regression to the mean” as confounding mechanism. Moreover, a selection bias in children who got omega-3-fatty acid supplementation lies on the fact that their parents self-select themselves for the treatment, prove a higher motivation for the therapy and remain susceptible to a placebo effect. This means that the results of our analysis are hypothesis generating and deserve further exploration in a randomized control study. 

Yours sincerely

Prof. Dr. Reiner Buchhorn and Dr. Christian Willaschek

1. Still AM, Raatikainen P, Ylitalo A, Kauma H, Ikäheimo M, Antero Kesäniemi Y, and Huikuri HV. Prevalence, characteristics and natural course of inappropriate sinus tachycardia. Europace 7: 104-112, 2005.

---

## [Decision Letter · Decision Letter 2]

11 Aug 2020

Diagnosis and management of an inappropriate sinus tachycardia in adolescence based upon a Holter ECG: A retrospective analysis of 479 patients

PONE-D-20-09485R2

Dear Dr. Willaschek,

We’re pleased to inform you that your manuscript has been judged scientifically suitable for publication and will be formally accepted for publication once it meets all outstanding technical requirements.

Kind regards,

Antonio Cannatà

Academic Editor

PLOS ONE

Additional Editor Comments (optional):

Reviewers' comments:

Reviewer's Responses to Questions

**Comments to the Author**

1. If the authors have adequately addressed your comments raised in a previous round of review and you feel that this manuscript is now acceptable for publication, you may indicate that here to bypass the “Comments to the Author” section, enter your conflict of interest statement in the “Confidential to Editor” section, and submit your "Accept" recommendation.

Reviewer #1: All comments have been addressed

2. Is the manuscript technically sound, and do the data support the conclusions?

Reviewer #1: Yes

3. Has the statistical analysis been performed appropriately and rigorously? 

Reviewer #1: Yes

4. Have the authors made all data underlying the findings in their manuscript fully available?

Reviewer #1: Yes

5. Is the manuscript presented in an intelligible fashion and written in standard English?

Reviewer #1: Yes

6. Review Comments to the Author

Reviewer #1: This is a revised manuscript about a retrospective study evaluating 479 children for the association of inappropriate sinus tachycardia (IST, n=79) with heart rate variability (HRV) in Holter-ECG. Healthy controls (n=66) are compared to 8 different groups with attention deficit disorder (ADD, n=86), obesity (n=130), short stature (n=15), congenital heart disease (n=55), anorexia nervosa (n=34), constitutional thinness (n=20), hypertension (n=53) or pre-hypertension (n=13). IST risk was higher in adolescents with attention deficit disorder (OR=3.5,p<0.001), hypertension (OR=2.1,p=0.02) and pre-hypertension, but did not reach statistical significance in other groups. IST was associated with decreased HRV and omega-3-fatty acid supplementation was associated with decreased HR and increased HRV in adolescents with IST.

The authors are commended for the revision. Their efforts are highly appreciated and the manuscript has significantly improved. Thank you and congratulations!

7. PLOS authors have the option to publish the peer review history of their article (what does this mean?). If published, this will include your full peer review and any attached files.

Reviewer #1: No

---

## [Editor Report · Acceptance letter]

17 Aug 2020

PONE-D-20-09485R2 

Diagnosis and management of an inappropriate sinus tachycardia in adolescence based upon a Holter ECG: A retrospective analysis of 479 patients 

Dear Dr. Willaschek:

I'm pleased to inform you that your manuscript has been deemed suitable for publication in PLOS ONE. Congratulations! Your manuscript is now with our production department. 

Kind regards, 

on behalf of

Dr. Antonio Cannatà 

Academic Editor

PLOS ONE